# A Reversible Protonic Ceramic Cell with Symmetrically Designed Pr_2_NiO_4+δ_-Based Electrodes: Fabrication and Electrochemical Features

**DOI:** 10.3390/ma12010118

**Published:** 2018-12-31

**Authors:** Artem Tarutin, Julia Lyagaeva, Andrey Farlenkov, Sergey Plaksin, Gennady Vdovin, Anatoly Demin, Dmitry Medvedev

**Affiliations:** 1Laboratory of Electrochemical Devices Based on Solid Oxide Proton Electrolytes, Institute of High Temperature Electrochemistry, Yekaterinburg 620137, Russia; vanomass333@gmail.com (A.T.); yulia.lyagaeva@ya.ru (J.L.); vdovin@ihte.uran.ru (G.V.); a.demin@ihte.uran.ru (A.D.); 2 Institute of New Materials and Technologies, Ural Federal University, Yekaterinburg 620002, Russia; 3 Institute of Chemical Engineering, Ural Federal University, Yekaterinburg 620002, Russia; a.farlenkov@yandex.ru; 4 Laboratory of Solid State Oxide Fuel Cells, Institute of High Temperature Electrochemistry, Yekaterinburg 620137, Russia; plaksin@ihte.uran.ru; 5 Graduate School of Economics and Management, Ural Federal University, Yekaterinburg 620002, Russia

**Keywords:** PCFCs/PCECs, Ruddlesden-Popper phases, symmetrical cells, proton-conducting electrolytes

## Abstract

Reversible protonic ceramic cells (rPCCs) combine two different operation regimes, fuel cell and electrolysis cell modes, which allow reversible chemical-to-electrical energy conversion at reduced temperatures with high efficiency and performance. Here we present novel technological and materials science approaches, enabling a rPCC with symmetrical functional electrodes to be prepared using a single sintering step. The response of the cell fabricated on the basis of P–N–BCZD|BCZD|PBN–BCZD (where BCZD = BaCe_0.5_Zr_0.3_Dy_0.2_O_3−δ_, PBN = Pr_1.9_Ba_0.1_NiO_4+δ_, P = Pr_2_O_3_, N = Ni) is studied at different temperatures and water vapor partial pressures (pH_2_O) by means of volt-ampere measurements, electrochemical impedance spectroscopy and distribution of relaxation times analyses. The obtained results demonstrate that symmetrical electrodes exhibit classical mixed-ionic/electronic conducting behavior with no hydration capability at 750 °C; therefore, increasing the pH_2_O values in both reducing and oxidizing atmospheres leads to some deterioration of their electrochemical activity. At the same time, the electrolytic properties of the BCZD membrane are improved, positively affecting the rPCC’s efficiency. The electrolysis cell mode of the rPCC is found to be more appropriate than the fuel cell mode under highly humidified atmospheres, since its improved performance is determined by the ohmic resistance, which decreases with pH_2_O increasing.

## 1. Introduction

Solid oxide systems with predominant protonic transport are considered as advanced applied materials for the energy sector. A particular interest is associated with their utilization as proton-conducting electrolytes in protonic ceramic cells (rPCCs), reversible solid oxide cells that are used to convert different types of energy with high efficiency and no harmful impact [1,2,3,4,5]. Compared with conventional solid oxide cells with oxygen-ionic electrolytes, rPCCs are able to operate at reduced temperatures due to their high proton mobility and low activation energy [6,7,8]. These advantages may feasibly lead to their commercialization in the near future [9]. Therefore, carrying out the necessary applied and fundamental research in this field, as well as developing new approaches for the optimization of the relevant technological and electrochemical processes, are among highly relevant current trends. 

Recently, many efforts have been made to simplify technological processes involved in the fabrication of solid oxide devices based on oxygen-ionic or proton-conducting electrolytes [10,11]. One of these efforts consists in designing electrochemical cells having symmetrical electrodes [12] as an efficient strategy for reducing fabrication costs. Here, the economic benefit lies in minimizing the number of functional materials used. Moreover, this strategy can help to resolve problems associated with thermal incompatibility and electrochemical degradation if the latter has a reversible nature [13,14]. However, materials for symmetrical electrodes preparation must satisfy a number of requirements, such as excellent electrochemical activity in both reduced (Red) and oxidized (Ox) conditions, as well as appropriate phase and thermal behavior under RedOx cycling. As a rule, only a limited series of materials can be used for this purpose, including Ti-, Fe-, Cr- and Mn-based oxides with simple or double perovskite structures [12].

Although the utilization of symmetrical solid oxide fuel cells based on oxygen-ionic electrolytes has been intensively studied in the past few years [13,14,15,16,17], the application of this strategy to PCCs is only at the beginning of its development. In the present work, we propose to use a Pr_2_NiO_4+δ_-based oxide in symmetrical functional layers for a rPCC. In contrast to the previously mentioned simple or double perovskite, a layered structure of praseodymium nickelate is substituted under reducing atmospheres to complete Ni reduction and formation of a Ni–Pr_2_O_3_ cermet (Figure 1) with good electrocatalytic properties [18,19]. Along with the symmetrical electrode application and reversible operation mode, the close thermal expansion coefficients (TECs) of Pr_2_NiO_4+δ_- and Ba(Ce,Zr)O_3_-based materials [20,21] allow a one-step sintering procedure to be used. According to the literature analysis, a single temperature processing step is a highly attractive approach for reducing the fabrication costs; in particular, this strategy has recently been adopted for the production of PCCs [8,22,23]. However, in these works the anode and cathode layers consisted of different functional materials, which can cause a mechanical misbalance leading to the deformation of whole cells following their sintering. Utilizing the same component for both functional electrode layers minimizes the possible strain, representing significant benefits in technological aspects, as well as the in terms of the quality of the target product.

## 2. Materials and Methods 

### 2.1. Preparation of Materials

The protonic ceramic cell (PCC) was fabricated from three functional materials, including BaCe_0.5_Zr_0.3_Dy_0.2_O_3−δ_ (BCZD) as a proton-conducting electrolyte layer (EL), its mixture with nickel oxide as a substrate for supporting fuel electrode layer (SFEL) and Pr_1.9_Ba_0.1_NiO_4+δ_ (PBN) as a basis for functional oxygen (FOEL) and functional fuel (FFEL) electrode layers.

The BCZD and BPN powders were obtained using the citrate-nitrate synthesis method described in detail in our previous works [24,25].

### 2.2. Characterization of Materials

The phase structure of the individual materials (BCZD, PBN) and their mixture (1:1 at ratio) calcined at 1350 °C for 3 h was studied by X-ray diffraction analysis (Rigaku D/MAX-2200VL [26]). The scans were performed under CuK_α1_ radiation between 20° and 80° with a scan step of 0.02° and a scan rate of 3° min^−1^.

The morphology of the ceramic samples and multilayered cell was studied using scanning electron microscopy (SEM) and energy dispersive X-ray (EDX) analyses on a Tescan Mira 3 LMU microscope with an Oxford Instruments INCA Energy X-MAX 80 spectrometer [26]. 

Thermogravimetric (TG) technique (Netzsch STA 449 F3 Jupiter) was employed to study reduction behavior of the PBN powder.

Conductivity measurements were carried out in air and wet H_2_ atmospheres using a convenient 4-probe DC current method (Zirconia-318 measurement station).

### 2.3. Fabrication of the PCC

The single-phase BCZD was mixed with NiO and starch (pore former) in a weight ratio of 3:2:1 to prepare the powder for the SFEL, while BCZD was mixed with PBN and pore former in a weight ratio of 4:1:1 to be used as FOEL and FFEL. The mixing stages were performed using a Fritsch Pulverisette 7 planetary ball mill with the following conditions: zirconia milling balls, acetone media, 500 rpm for 0.5 h. The corresponding powders were mixed with an organic binder (butadiene rubber in acetone/benzene solvent) and rolled used a Durtson rolling mill to fabricate the functional films having the required thicknesses. These films were then co-rolled with each other (tape-calendering method), adjusted by ~30 µm for the raw EL, ~20 µm for the raw FOEL and FFEL and ~800 µm for the raw SFEL. The green multilayered NiO–BCZD|PBN–BCZD|BCZD|PBN–BCZD structure was slowly (1 °C min^−1^) heated up to 900 °C to decompose and remove organic residue, then heated further to 1350 °C at a heating rate of 5 °C min^−1^, sintered at this temperature for 3 h and cooled to room temperature at a cooling rate of 5 °C min^−1^.

### 2.4. Characterization of the PCC

The fabricated PCC with an effective electrode area of 0.21 cm^2^ was characterized at a temperature range of 600–750 °C under reversible operation (bias changes from 0.1 to 1.5 V with a step of 25 mV). The volt-ampere dependences and impedance spectra were obtained using a complex of Amel 2550 potentiostat/galvanostat and Materials M520 frequency response analyzer. To evaluate the electrode response, hydrogen as a fuel and air as an oxidant were humidified to varying degrees. The target water vapor partial pressure values were set by passing the corresponding atmospheres through a water bubbler heated to certain temperatures. The impedance spectra were obtained across a frequency range of 10^−2^–10^5^ Hz at a perturbation voltage of 25 mV and were then analyzed utilizing the methods of equivalent circuits (Zview software) and distribution of relaxation times (DRT, DRTtools core of the Matlab software [27]).

## 3. Results and Discussion

### 3.1. Pr_1.9_Ba_0.1_NiO_4+δ_ Functionality

To assess the applied prospect of PBN as electrodes for PCCs, its properties were comprehensively studied (Figure 2). 

This material is characterized by a high chemical compatibility with the proton-conducting electrolytes based on Ba(Ce,Zr)O_3_ (Figure 2a), since the high-temperature treatment results in maintaining the basic structures for both phases, although a small amount of NiO exists at the same time (this impurity phase is also visible at the interface regions, see the Ni-element distribution map on Figure 3c). According to our best knowledge, we have used the highest temperature (1350 °C), at which chemical interactions of cerate-zirconates and nickelates were studied. Such excellent results in phase stability can be explained by the fact that Pr_2_NiO_4+δ_ is doped with barium, that compensates the Ba-concentration difference between two components and, therefore, diminishes a degree of chemical interaction [28,29]. 

As it is shown in Figure 2b, the PBN material in a powder state starts to reduce at 300 °C, when interstitial (and a part of lattice-site) oxygen is gradually removed [30]; this nickelate is almost completely decomposed at temperatures above 600 °C, which is in line with the following simplified reaction: Pr_2_NiO_4+δ_→Pr_2_O_3_ + Ni, see Figure 1.

Conductivity of the as-prepared ceramic sample in air atmosphere differs from one of the reduced samples in wet hydrogen by more than four orders of magnitude (Figure 2d). It can be concluded that the reduced sample represents a mixture of (Pr,Ba)_2_O_3_ and Ni with a mole ratio of 2:1. In other words, the total conductivity is determined by oxide phases, since no continuous high conductive metallic framework is formed under such a reduction. Nevertheless, Ni-particles appear during the exsolution procedure (Figure 2c), which is found to be a remarkable factor in electrode processes improvement [31]. According to Figure 2c, a certain part of Pr_2_O_3_ is transformed to Pr(OH)_3_, but this hydroxide is formed at relatively low temperatures only (under cooling of the sample in static air), when Pr_2_O_3_ chemisorbs steam of air. This is confirmed by the fact that the total weight change of PBN under full reduction is equal to 95.2% (δ_@RT_ = 0.17, 4 + δ_@1000 °C_ = 2.95; see Figure 2b) corresponding with the formation of 0.95 mole of Pr_2_O_3_ and 0.1 mole of BaO at 1000 °C. Otherwise, if Pr(OH)_3_ was formed during the reduction procedure, the overall weight change should amount to 96.1%. This level corresponds to δ_@RT_ = −0.07 (4 + δ_@1000 °C_ = 2.95), which is in disagreement with that reached for Pr_2_NiO_4+δ_, which has a close composition, δ = 4.23–4.25 [32,33].

### 3.2. Microstructural Features 

The PCC fabricated using the one-step sintering procedure shows a well-organized, multi-layered structure without any visible deformation, material delamination or cracks (Figure 3a). The obtained results can be explained by the excellent thermal compatibility of the Ln_2_NiO_4+δ_-based (Ln = La, Pr, Nd) materials with Ba(Ce,Zr)O_3_ proton-conducting electrolytes, especially the low chemical expansion in contrast to that of many electrode materials having perovskite-related structures [34,35]. 

The resulting thickness of the FOEL, EL and FFEL are estimated to be about 15, 25 and 17 µm, respectively (Figure 3b), while the total PCC thickness is equal to 700 µm. The porosity of SFEL evaluated using ImageJ software amounts to 40 ± 5 vol.%; the porosity of the functional electrodes does not exceed 20% (measured on the individually prepared pellets with the same composition, BCZD:PBN:starch = 4:1:1), indicating their strong sintering behavior despite 20 wt.% of pore former used.

According to EDX analysis (Figure 3c), elements are evenly distributed and do not show significant interdiffusion, forming clearly separated interphase boundaries. This also supports the conclusion regarding the chemical compatibility of materials used, at least at a sintering temperature of 1350 °C.

### 3.3. Volt-Ampere Dependences and Related Properties

The operability of the reversible PCC is shown in Figure 4a. In the current-free mode, this cell generates open circuit voltages (OCV) of 1.076, 1.051, 1.006 and 0.979 V at 600, 650, 700 and 750 °C, respectively. These values are somewhat lower than those theoretically predicted, which amount to 1.135, 1.126, 1.118 and 1.109 V, respectively. Since this cell maintains gas-impenetrability (measured at room temperature under 10^−3^ atm/1 atm of total gas pressures gradient), the most likely reason for the observed differences is non-ionic conduction of the BCZD electrolyte. Indeed, both BaCeO_3_- and BaZrO_3_-based materials demonstrate meaningful electron conductivity in oxidizing conditions at high temperatures [36,37]. Regarding the BCZD electrolyte, its ionic transference numbers (t_i_) estimated as a ratio of its ionic conductivity measured in wet H_2_ to the total conductivity measured in wet air only reached 0.78, 0.62, 0.50 and 0.44 at 600, 650, 700 and 750 °C [38]. It should be noted that these t_i_ values were determined for the samples in system with unseparated gas space, when all the samples’ sides were in contact with the same atmosphere. In the case of PCC, the electrolyte membrane separates two gas spaces. Therefore, one subsurface of the membrane exhibits predominant proton transport, while another one features mixed ionic-electronic transport. Therefore, the resulting (or average) ionic transference numbers, t_i,av_, should be much higher than t_i_. The exact t_i,av_ values under OCV conditions can be calculated as follows [39]:(1)$$t_{i,\mathrm{av}}=1-\frac{R_{O}}{R_{O}+R_{p}}\left(1-\frac{E_{\mathrm{meas}}}{E}\right),$$
where, R_O_ and R_p_ are the ohmic and polarization resistances of the PCC, while E_meas_ and E are the measured and theoretically predicted potentials. As can be seen, this equation includes the parameters related to the resistances of the functional materials, which can be separated using the impedance spectroscopy method. The separation procedure will be described in Section 3.4 and the calculation of t_i,av_ values shown in Section 3.6.

The PCC yields such maximal power densities (P_max_) as 305, 360, 395 and 430 mW cm^−2^ at 600, 650, 700 and 750 °C, respectively (Figure 4b,d). In electrolysis cell mode of operation, the maximal current densities reach about 640, 780, 950 and 1070 mA cm^−2^ (U = 1.5 V) at the corresponding temperatures and about 295, 405, 535 and 690 mA cm^−2^ under conditions close to the thermoneutral mode (U_TN_ ≈ 1.3 V, Figure 4c,d). This mode is determined by the thermodynamic parameter of the resulting reaction occurring in the rPCC (Equations (2) and (3)) and corresponds to the conditions when the cell is in thermal equilibrium. More precisely, the rPCC consumes heat when the bias is lower than U_TN_ and, conversely, produces heat when the bias exceeds U_TN_.
(2)$$H_{2}+1/2O_{2}⇄H_{2}O,$$
(3)$$U_{TN}=\frac{-\Delta H}{zF}.$$

Here, ΔH is the molar enthalpy of reaction (2), z is the number of participating electrons (z = 2) and F is the Faraday constant.

### 3.4. Analysis of Impedance Data

First, the PCC’s functionality was characterized under OCV mode using EIS analysis (Figure 5). As can been seen from these data, all the obtained impedance spectra consist of two clearly separated arcs, corresponding to low- and high-frequency processes. In order to correlate these partial processes with the corresponding resistances, the experimental results were fitted by the model lines originated from the used L—R_O_—(R_1_Q_1_)—(R_2_Q_2_) equivalent circuit scheme. Here, L is the inductance associated with cables, wires and their junctions, R_O_ is the ohmic resistance of the electrolyte membrane, R_1_ and R_2_ are the resistances of low-(I) and high-(II) processes, Q_1_ and Q_2_ are the corresponding constant phase elements. Two fitting results (presented in Figure 5 as examples) confirm a good agreement between experimental and model data, implying the success of the used scheme.

According to the fitting procedure, the R_O_, R_1_, R_2_ parameters, along with the polarization resistance of the electrodes (R_p_ = R_1_ + R_2_) and total resistance of the PCC (R_total_ = R_O_ + R_p_), were successfully determined, as shown in Figure 6. With increasing temperature, the total polarization resistance of the PCC decreases from 0.90 to 0.52 Ω cm^2^ (Figure 6a); at the same time, this resistance is determined by the ohmic component (Figure 6b), the contribution of which varies between 57 and 86%. Two factors contribute to this result: the rather high thickness of the electrolyte used (25 µm) and the excellent electrochemical properties of the electrodes, despite their fairly low porosity. The total polarization resistance of the electrodes decreases from 0.39 Ω cm^2^ at 600 °C to 0.07 Ω cm^2^ at 750 °C and is regulated by the R_2_ level, which contributes to amounts to 91% and 71%, respectively.

All the constituent resistances have a thermo-activated nature with different activation energies, E_a_ (see Appendix A, Figure A1). The Process II exhibits the highest E_a_ value, indicating a strong correlation with temperature; conversely, the E_a_ level of Process I is lower than that of Process II by ~3 times. The E_a_ level of the PCC’s total resistance is quite low (0.37 eV), since it is regulated by the predominant influence of the ohmic resistance with the lowest E_a_. Two important conclusions are supported by the obtained data: (1)The low E_a_ for the ohmic resistance is an indirect evidence of proton behavior, since protons migrate much more easily than massive oxygen-ions [40,41];(2)The slight temperature behavior of the total resistance of the PCC is a characteristic feature of the cells, corresponding to the condition of R_p_ < R_O_. Therefore, their output parameters (as shown in Figure 4d) also change slightly with temperature variation.

In order to evaluate the prospect of utilizing the Pr_2_NiO_4+δ_-based material as symmetrical electrodes, Processes I and II were thoroughly analyzed by calculating the capacitance/frequency values and utilizing the DRT method.

The values for the capacitance and frequency characteristics (at the top of the arcs) were estimated from the impedance spectra fitting as follows:(4)$$C_{j}={\left(R_{j}\cdot Q_{j}\right)}^{1/n_{j}}\cdot R_{j}^{-1},$$
(5)$$f_{j}={\left(R_{j}\cdot Q_{j}\right)}^{-1/n_{j}}\cdot {\left(2\pi \right)}^{-1},$$
where j = 1 or 2, n_j_ is the exponent factor [42,43].

Process I semicircles are characterized by the characteristic capacitances of 3.6·× 10^−3^–6.4·× 10^−3^ F cm^−2^ and characteristic frequencies of 4 × 10^2^–6 × 10^2^ Hz; these values reach 8.5 × 10^−1^–9.0 × 10^−1^ F cm^−2^ and 4 × 10^−1^–4 × 10^1^ Hz, respectively, for the semicircles of Process II. These results are also supported by the DRT data (Figure 7). As can be seen, Processes I and II correspond to the medium- and low-frequency stages, respectively. In detail, the first rate-determining step can be attributed either to the surface charge-transfer phenomenon [44] or to ionic diffusion at the electrode [45] with direct participation of proton charge carriers due to having a low E_a_ value (Figure A1). Although low values are achieved for this stage at the estimated frequencies, it should be noted that the electrodes (at least, the oxygen one) represent quite a dense structure for an electrolytic component, which might provide proton transportation. Due to high E_a_ and C_2_ values, the second rate-determining (and dominating) step corresponds to sluggish gas-diffusion and adsorption of electrochemically-active components [46,47], which correlates with the mentioned low porosity of the electrodes.

Along with these two steps, the high-resolution DRT method [48,49] gives an additional peak around tens of Hz (see inset in Figure 7). The partial polarization resistance corresponding to this peak (R1’) does not exceed 0.005 and 0.002 Ω cm^2^ (below 1.5% of R_p_) at 600 and 650 °C, respectively; therefore, it cannot be resolved under convenient impedance data analysis by equivalent circuits. Considering its mediate frequencies and strong dependence on temperature, such a peak can likely be associated with the dissociation of adsorbed molecules [46,47].

Summarizing, the symmetrically-formed electrodes yield a promising performance of the PCC regardless of their low-porous microstructure. Moreover, the obtained results are in line with characteristics for similar PCC electrolyte and oxygen electrode materials (Table 1, [50,51,52,53]).

### 3.5. Effect of Air and Hydrogen Humidification

It is well-known that water vapor partial pressure is a parameter determining proton transport in proton-conducting electrolyte membranes [54]. From the viewpoint of the bulk structure of such membranes, humidification of both atmospheres is favorable, since it results in a decrease both of oxygen vacancies and hole concentrations:(6)$$V_{O}^{••}+O_{O}^{x}+H_{2}O⇄2OH_{O}^{•},$$
(7)$$V_{O}^{••}+1/2O_{2}⇄2h^{•}+O_{O}^{x},$$
and, correspondingly, improved proton transport. At the same time, air humidification is considered to be a more effective and easy way of suppressing some of the undesirable electronic conductivity of cerates and zirconates [55,56].

From the perspective of thermodynamic features, the maximal achievable electrical potential difference of a PCC (E or OCV) decreases with gas humidification, which follows from the corresponding decrease in partial pressure gradients [57,58]:(8)$$E=t_{i,av}\frac{RT}{4F}\ln\left(\frac{p\prime O_{2}}{p\prime \prime O_{2}}\right)+t_{H,av}\frac{RT}{2F}\ln\left(\frac{p\prime \prime H_{2}O}{p\prime H_{2}O}\right)=t_{i,av}E_{O}+t_{H,av}E_{H_{2}O},$$
where p’O_2_ and p’’O_2_ are the oxygen partial pressures in reducing and oxidizing atmospheres, p’H_2_O, p’’H_2_O are the water vapor partial pressures, E_O_ and $E_{H_{2}O}$ are the electrical potential differences of oxygen- and steam-concentration cells, t_H,av_ is the average proton transference number. Such a decrease negatively affects the performance of the cells operated in fuel cell mode because, in order to obtain high power densities, high OCVs are necessary [59]:(9)$$P_{\max}=\frac{E^{2}}{4R_{total}}.$$

Conversely, the lowest possible OCVs are needed for cells operated in electrolysis cell mode if these OCVs are not associated with significant electron transport of the electrolytes or imperfect system gas-tightness. In detail, a current density will be higher at a certain voltage value (U), while a difference of U—E will be higher (or E will be lower).

In order to check the pH_2_O effect, both atmospheres were consequentially humidified: first, air atmosphere and then hydrogen atmosphere. Moreover, this allows the response of each electrode to be revealed and even their contributions to the total polarization resistance, R_p_, to be estimated.

Figure 8 shows the main electrochemical characteristics of the PCC obtained under isothermal conditions with gradual increase of pH_2_O. As can be seen from these data, the OCVs drop from 0.979 to 0.899 V, the maximal power density decreases from 430 to 290 mW cm^−2^, while the maximal achievable hydrogen flux density increases from 4.4 to 5.1 ml min^−1^ cm^−2^, respectively, when p’’H_2_O increases from 0.03 to 0.5 atm. This is in complete agreement with the abovementioned thermodynamic predictions.

EIS and DRT analyses were further utilized to reveal the main tendencies in R_o_ and R_p_ changes and their effects on the PCC’s performance.

Air humidification virtually does not change the spectra’s shape (Figure A2); they, as well as the original spectra, can accurately be described by an equivalent circuit scheme with two RQ-elements. On the base of DRT data (Figure A3), the distribution function consists of three well-separated peaks at high p’’H_2_O values, two of which merge at lower p’’H_2_O values. The total polarization resistance of the electrodes amounts 0.07, 0.08, 0.10 and 0.12 Ω cm^2^ at 0.03, 0.10, 0.30 and 0.50 atm of p’’H_2_O. At the same time, the partial resistance components vary differently (Figure A4): (1)The absolute value of R1 is equal to 0.02 Ω cm^2^, but its contribution as part of R_p_ decreases from 30 to 17%; (2)The contribution of R1’ in R_p_ does not exceed 4.5% or 0.004 Ω cm^2^ in absolute units;(3)The contribution of R_2_ increases from 70 to 80%, remaining the dominant parameter in the electrode performance.

All the mentioned components are sensitive towards air humidification. Therefore, these stages are primarily associated with oxygen electrode behavior.

Returning to the data of Figure 8, it can be stated that performance of the PCC is regulated by the electrode activity, which dominates under OCV and fuel cell modes of operation. When the bias exceeds the OCV level, the electrode resistance drops rapidly [60]; in this case, an improvement in PCC’s performance is related with a lower R_O_ as a result of achieving excellent proton conductivity in highly humid conditions. Making a preliminary conclusion, it can be noted that the Pr_2_NiO_4+δ_-based electrodes operate as a classical mixed oxygen-ionic/electronic conductor with no evidence of proton transportation revealed in previously published works [51,61]. However, this might be explained by the relatively high measured temperature (750 °C) leading to the insignificant water uptake capability of nickelates.

Finally, the PCC was tested depending on temperature under 50%H_2_O/H_2_—50%H_2_O/air conditions, corresponding to both highly-moisturized gases (Figure 9). As listed in Table A1, further hydrogen humidification from 0.03 to 0.5 atm has little effect on both maximal power density (decreases by ~3%) and the hydrogen evolution rate (increases by ~6%). Therefore, differences in output parameters obtained at p’H_2_O = p’’H_2_O = 0.03 atm (condition 1) and p’H_2_O = p’’H_2_O = 0.5 atm (condition 2) are mainly attributed with the pH_2_O variation in air atmosphere. Such differences amount to −37% of P_max_ and +13% of jH_2_ at the same comparison temperature as shown in Figure 9d. The obtained data are also in agreement with thermodynamic predictions, in particular with OCVs, which reach 0.991, 0.951, 0.912 and 0.874 V at 600, 650, 700 and 750 °C, respectively.

With increasing pH_2_O in hydrogen atmosphere, the impedance spectra cannot be described by the used equivalent circuit schemes, implying the appearance of additional rate-determining steps. These steps might be attributed either to fuel electrode processes or even to those taking place at the oxygen electrode. The latter is realized due to the fact that a change in a potential-determined parameter from the one side of an electrolyte membrane results in a redistribution of the overall potential and internal (ionic and electronic) currents, which can in turn affect the electrode process at the other side of the same membrane. As indicated in Figure A5, hydrogen humidification (when the oxidizing composition is unchanged) leads to the formation of a new distribution function, consisting in different number of peaks, as well as their intensity and displacement. Such a distribution function becomes more complicated with decreasing temperature (Figure 10): here, at least five independent peaks and corresponding processes can be distinguished. 

Due to the already mentioned peculiarities, the R_O_ values were determined as a high-frequency intercept of spectra with the x-axis, while the R_p_ values were separately estimated as the total area bounded by a γ(τ) function. For 50%H_2_O/H_2_—50%H_2_O/air and OCV conditions, R_O_ is equal to 0.51, 0.48, 0.45 and 0.42 Ω cm^2^, whereas R_p_ is equal to 0.99, 0.46, 0.21 and 0.14 Ω cm^2^ at 600, 650, 700 and 750 °C, respectively. Comparing 50%H_2_O/H_2_—50%H_2_O/air and 3%H_2_O/H_2_—50%H_2_O/air conditions at 750 °C, the R_O_ parameter is virtually unchanged, showing the possibility of full hydration of the proton-conducting electrolyte. At the same time, R_p_ increases from 0.12 to 0.14 Ω cm^2^. Such an increment of R_p_ reaches 0.99 Ω cm^2^ at 600 °C and 0.21 Ω cm^2^ at 700 °C for 50%H_2_O/H_2_—50%H_2_O/air conditions as against 0.39 Ω cm^2^ at 600 °C and 0.12 Ω cm^2^ at 700 °C for 3%H_2_O/H_2_—3%H_2_O/air conditions. Therefore, not only air humidification, but also hydrogen moisturization leads to higher R_p_ values and correspondingly lower performance characteristics. It should be noted that neither R_O_ nor R_p_ indicates the efficiency of the PCC, which can, however, be estimated on the basis of the electrolytic domain boundaries (absolute level and contribution of ionic conductivity).

### 3.6. Electrolytic Properties of the BCZD Membrane

In order to estimate the electrolytic properties of BCZD, which determines rPCCs’ efficiency [62,63,64], the t_i,av_ values were calculated (Equation (1)) using data on R_O_ and R_p_. According to Figure 11, the ionic transference numbers decrease naturally with increasing temperature due to an increase in the hole conductivity contribution. Nevertheless, the t_i,av_ parameter achieves quite a high level (0.90 at 750 °C) and, at the same time, can be further increased (up to 0.95 at the same temperature) via humidification of both gases. As a result of high saturation, oxygen vacancies are almost fully filled with steam, leading to a higher concentration of proton charge carriers (Equation (6)) and inhibition of hole charge carrier formation due to the lower concentration of free oxygen vacancies (Equation (7)). 

The average ionic conductivity of the electrolyte membranes, determined as
(10)$$\sigma_{i,av}=\frac{h}{R_{O}}\cdot t_{i,av},$$
should be considered as the most appropriate parameter (instead of total conductivity) related simultaneously with the performance (h/R_O_) as well as the efficiency (t_i,av_) of the electrochemical devices. For the present case, the average ionic conductivity of BCZD reaches 4.7 and 4.9 mS cm^−1^ at 600 and 700 °C under condition 1 and 4.9 and 5.4 mS cm^−1^, respectively, under condition 2. These belong to a range of the highest values reached for proton-conducting electrolytes (Table 2, [37,52,65,66,67,68,69,70,71,72,73,74,75,76,77,78,79]), which agrees with the independently measured comparison of ionic conductivities of Y- and Dy-doped Ba(Ce,Zr)O_3_ [80]. The abovementioned results allow the following important conclusions to be formulated:(1)The BCZD electrolyte forms the basis for the design of novel electrochemical cells with improved output parameters due to its higher ionic conductivity compared with those for the most-studied Y-containing cerate-zirconates;(2)Despite the negative electrochemical response of the electrodes to gas humidification, the average ionic transference and ionic conductivity values take the opposite direction, resulting in improved PCC efficiency. 

## 4. Conclusions

This work presents the results of fabrication and characterization of a reversible protonic ceramic cell having symmetrically-organized electrodes made of Pr_1.9_Ba_0.1_NiO_4+δ_ (PBN). Utilization of the same material as the functional fuel and oxygen electrode allows minimization of the thermo-chemical stress between the functional materials during high-temperature steps and even a reduction of these steps to one sintering stage, promising significant techno-economic benefits. The fabricated cell based on a 25 µm-thick BaCe_0.5_Zr_0.3_Dy_0.2_O_3−δ_ (BCZD) proton-conducting electrolyte demonstrated output characteristics as high as ~300 mW cm^−2^ at 600 °C in fuel cell mode and ~300 mA cm^−2^ in electrolysis cell mode at thermoneutral conditions. The cell was tested under conditions of varying humidity in order to evaluate electrode and electrolyte performance. It was found that the PBN–BCZD oxygen electrode determined the overall electrode performance at 750 °C, operating as a dual conducting (O^2−^/h^●^) system due to the negative electrochemical response to gas humidification. As a consequence of the high contribution of polarization resistance to the total resistance, the response of the rPCC’s performance was the same under open circuit voltage and fuel cell modes. On the other hand, the ohmic resistance and the average ionic transference numbers of the electrolyte membrane increased with increasing humidification, demonstrating its proton-conducting nature. Along with increased efficiency (as a result of improved electrolytic properties), the performance of the PCC was higher under humidification due to the total cell resistance being determined by the ohmic component. Although only medium performance was reached in this work (due to the rather thick electrolyte, 25 µm), the proposed strategies can effectively be used in future to resolve a number of technological issues.

## Figures and Tables

**Figure 1 materials-12-00118-f001:**
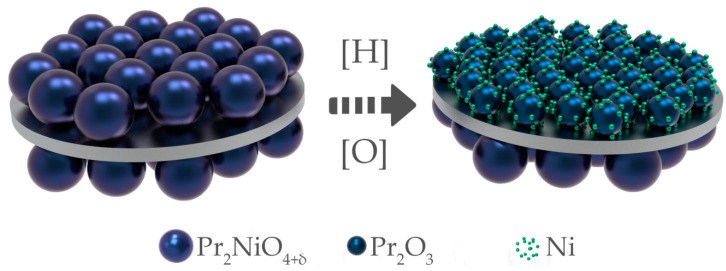
Principal scheme of a symmetrically designed PCC and Pr_2_NiO_4+δ_ reduction with the formation of a Ni-based cermet.

**Figure 2 materials-12-00118-f002:**
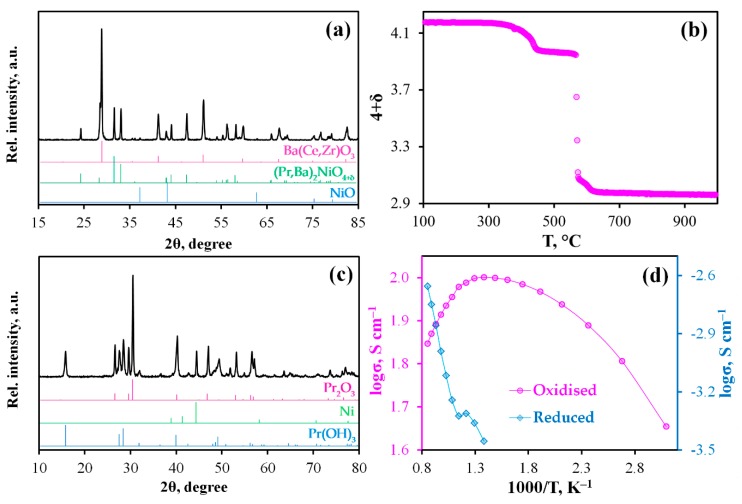
(**a**) XRD pattern of the BCZD – PBN mixture calcined at 1350 °C for 5 h; (**b**) TG curve obtained under reduction of the PBN powder in 50%H_2_/N_2_ atmosphere; (**c**) XRD pattern of the reduced product of the PBN material (after TG analysis); (**d**) Conductivity of the PBN samples in oxidizing and reducing atmospheres.

**Figure 3 materials-12-00118-f003:**
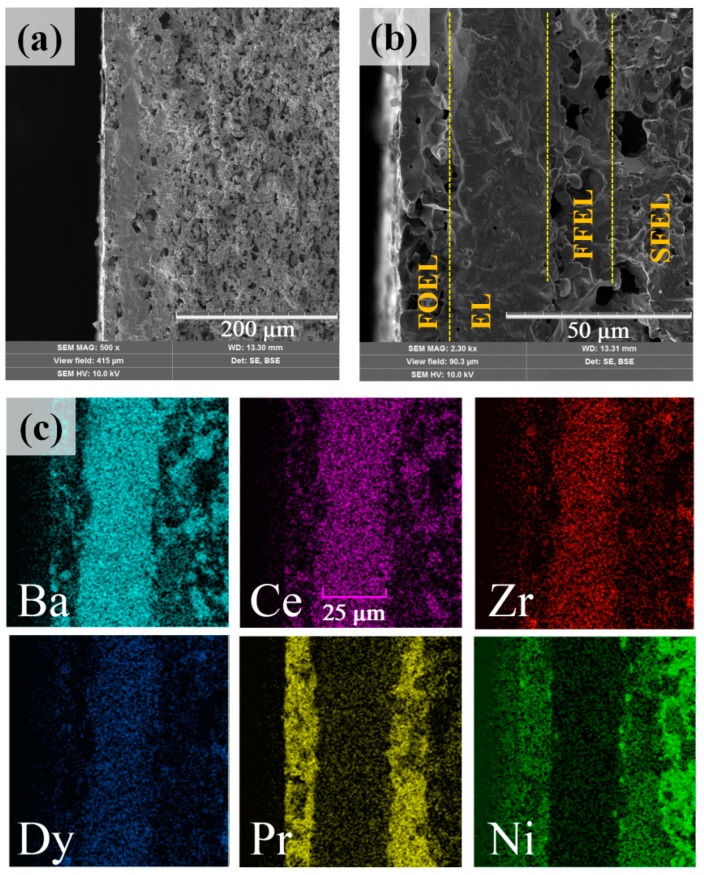
Cross-section images of the fabricated rPCC at different magnification (**a**,**b**) and maps of the elements distribution (**c**).

**Figure 4 materials-12-00118-f004:**
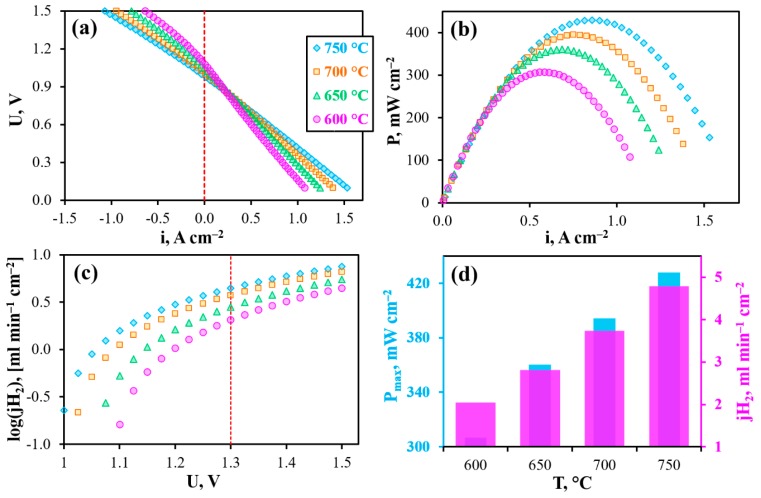
Reversible operation of the PCC at different temperatures under 3%H_2_O/H_2_–3%H_2_O/air conditions: volt-ampere curves (**a**), power density characteristics (**b**), maximal achievable hydrogen flux density (**c**), maximal power density and hydrogen flux density at U = 1.3 V depending on temperature (**d**).

**Figure 5 materials-12-00118-f005:**
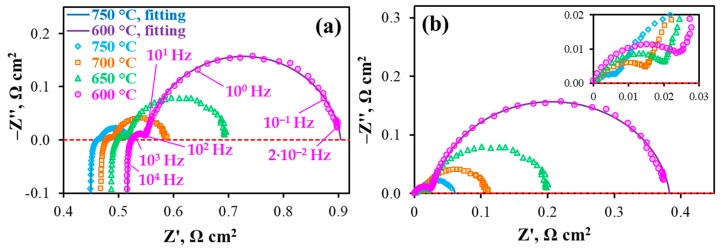
Impedance spectra of the PCC at different temperatures under 3%H_2_O/H_2_—3%H_2_O/air and OCV conditions: original spectra (**a**) and ones obtained after subtracting the ohmic resistance (**b**).

**Figure 6 materials-12-00118-f006:**
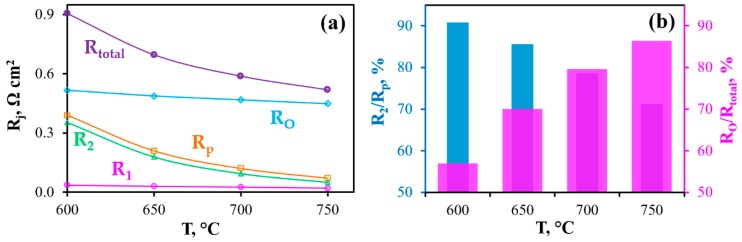
(**a**) Constituent resistances (R_j_) of the total resistance of the PCC depending on temperature; (**b**) Contributions of Process II in the total polarization resistance and the ohmic resistance in the total resistance of the PCC at different temperatures.

**Figure 7 materials-12-00118-f007:**
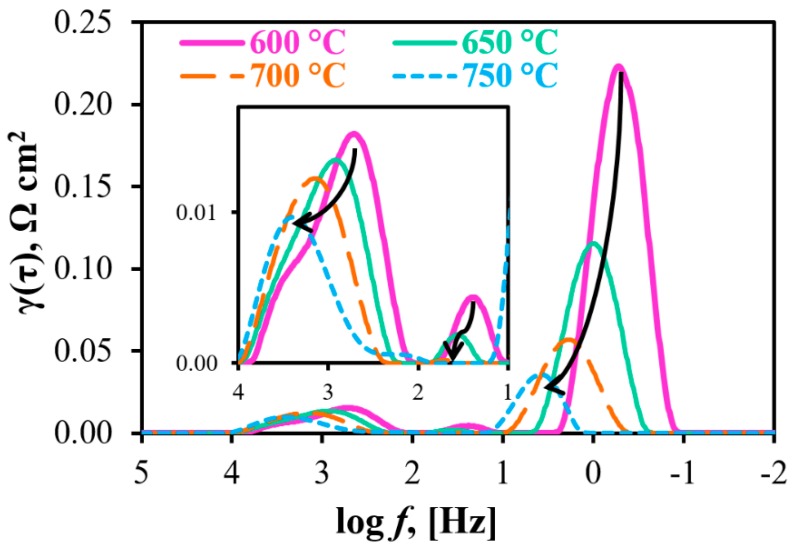
DRT results for the obtained impedance spectra (see details in Figure 5a).

**Figure 8 materials-12-00118-f008:**
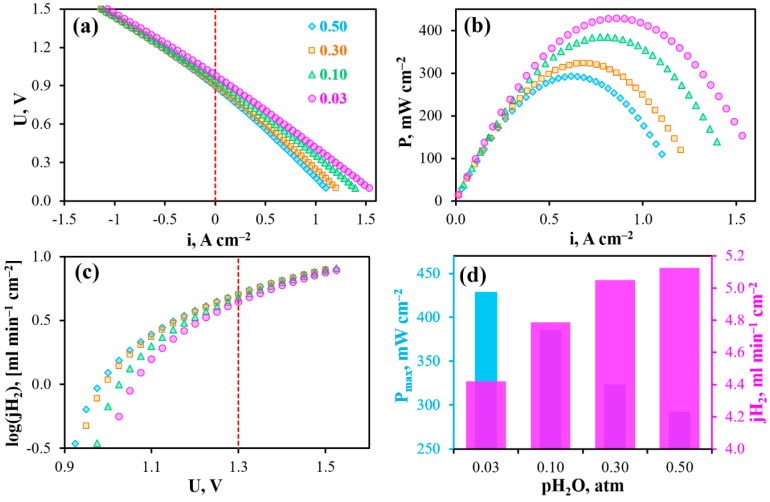
Reversible operation of the PCC at 750 °C depending on different pH_2_O in wet air with the constant fuel gas composition (3%H_2_O/H_2_): volt-ampere curves (**a**), power density characteristics (**b**), maximal achievable hydrogen flux density (**c**), maximal power density and hydrogen flux density at U = 1.3 V depending on pH_2_O (**d**).

**Figure 9 materials-12-00118-f009:**
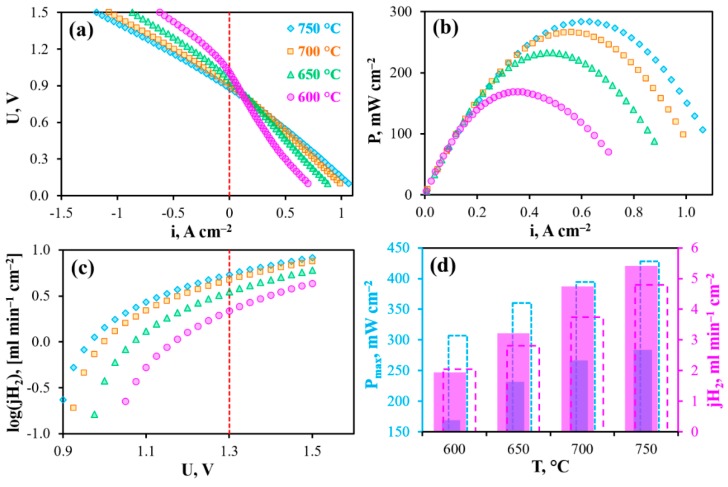
Reversible operation of the PCC at different temperatures under 50%H_2_O/H_2_—50%H_2_O/air conditions: volt-ampere curves (**a**), power density characteristics (**b**), maximal achievable hydrogen flux density (**c**), maximal power density and hydrogen flux density at U = 1.3 V depending on temperature compared with those (dashed columns) obtained under 3%H_2_O/H_2_—3%H_2_O/air conditions (**d**).

**Figure 10 materials-12-00118-f010:**
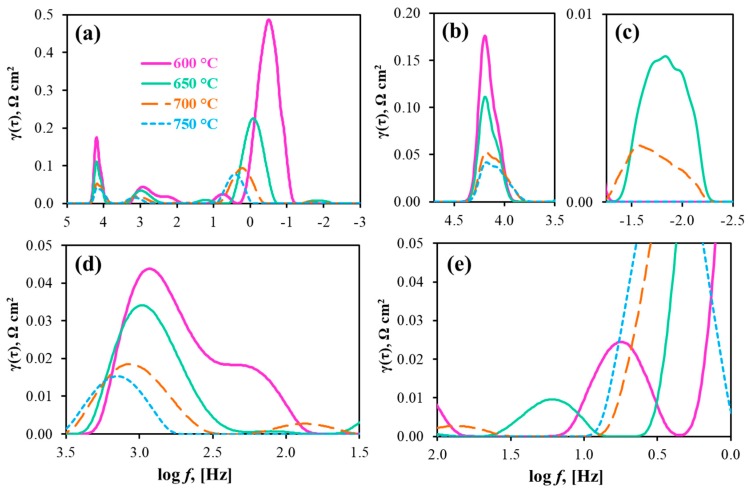
DRT results for the impedance spectra measured for the PCC at different temperatures under 50%H_2_O/H_2_—50%H_2_O/air and OCV conditions: general view (**a**) and its parts at different magnifications (**b**–**e**).

**Figure 11 materials-12-00118-f011:**
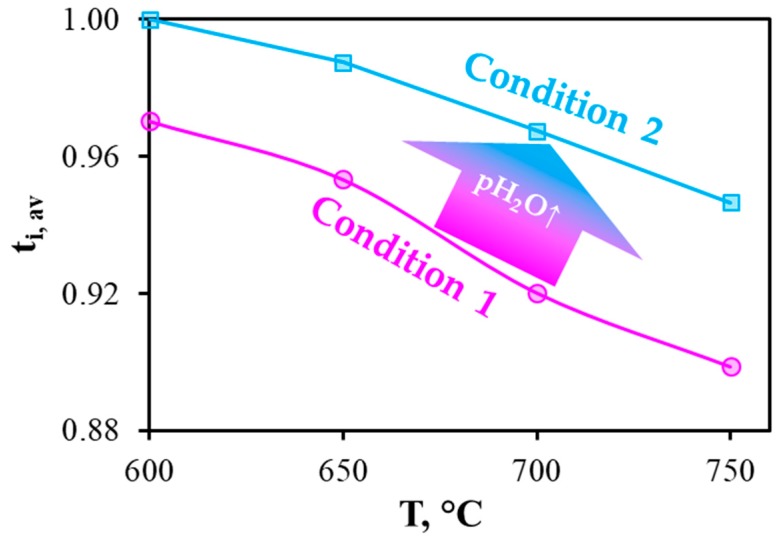
Temperature dependences of the average ionic transference numbers of the BCZD electrolyte membrane in the current-free mode of the rPCC under condition 1 (p’H_2_O = p’’H_2_O = 0.03 atm) and condition 2 (p’H_2_O = p’’H_2_O = 0.5 atm).

**Table 1 materials-12-00118-t001:** Polarization behavior of the Pr_2_NiO_4+δ_-based electrodes of PCCs under OCV mode of operation ^1^: T is the temperature, R_p_ is the total polarization of the electrodes.

Electrolyte	Electrode	T, °C	R_p_, Ω cm^2^	Ref.
BaCe_0.5_Zr_0.3_Dy_0.2_O_3−δ_ (BCZD)	Pr_1.9_Ba_0.1_NiO_4+δ_–BCZD	600	0.39	This work
		700	0.12	
BaCe_0.7_Zr_0.1_Y_0.2_O_3−δ_ (BCZY1)	Pr_1.8_Sr_0.2_NiO_4+δ_	600	2.17	[50]
		700	0.33	
BaCe_0.6_Zr_0.2_Y_0.2_O_3−δ_ (BCZY2)	Pr_2_NiO_4+δ_–BCZY2	600	0.21	[51]
		700	0.06	
BCZY1	(Pr_0.9_La_0.1_)_2_Ni_0.74_Cu_0.21_Nb_0.05_O_4+δ_– BCZY1 (infiltration)	600	0.32	[52]
		700	0.13	
BaCe_0.9_Y_0.1_O_3−δ_	Pr_2_NiO_4+δ_	600	0.80	[53]

^1^ Polarization resistance of the Ni-cermets is assumed to be much lower than that of Pr_2_NiO_4+δ_-based electrodes.

**Table 2 materials-12-00118-t002:** Electrolytic properties of proton-conducting membranes of PCCs under OCV conditions ^1^: h is the thickness, T is the temperature, E is the OCV, R_O_ and R_p_ are the ohmic and polarization resistances, t_i,av_ is the average ionic transference number, σ_av_ and σ_i,av_ are the average values of total and ionic conductivities of the electrolyte membranes.

Oxygen Electrode ^2^	Electrolyte ^3^	h, μm	T, °C	E, V	R_O_, Ω cm^2^	R_p_, Ω cm^2^	t_i,av_	σ_av_, mS cm^−1^	σ_i,av_, mS cm^−1^	Ref.
PBN	BCZD	25	600	1.076	0.52	0.39	0.97	4.8	4.7	This work
			700	1.006	0.47	0.12	0.92	5.3	4.9	
SFM	BZCY8	1200	700	0.96	38.09	3.59	0.87	3.2	2.7	[65]
SFM	BZCY8	20	600	1.03	0.80	1.47	0.97	2.5	2.4	[66]
			700	0.96	0.57	0.33	0.91	3.5	3.2	
PLNCN	BCZY1	12	600	0.99	0.33	0.32	0.94	3.6	3.4	[52]
			700	0.95	0.21	0.13	0.91	5.8	5.2	
PBC	BCZYYC	10	600	1.00	0.37	0.34	0.94	2.7	2.5	[67]
			700	0.99	0.26	0.12	0.92	3.8	3.5	
SSC–BCZY1^/^	BCZY1^/^	9	600	1.13	0.47	0.42	1.00	1.9	1.9	[68]
			700	1.07	0.31	0.10	0.97	2.9	2.8	
GBSC	BCZY1	20	600	1.01	0.65	0.39	0.93	3.1	2.8	[69]
			700	1.02	0.52	0.08	0.92	3.8	3.6	
NBFN	BCZY1	40	600	1.11	0.84	0.71	0.99	4.8	4.7	[70]
LSCF	BCZY4	30	600	1.07	0.72	0.24	0.96	4.2	3.9	[37]
			700	1.01	0.58	0.07	0.91	5.2	4.7	
	BCZY6	30	600	1.06	0.58	0.13	0.95	5.2	4.9	
			700	0.99	0.46	0.04	0.90	6.5	5.8	
	BCZY7	30	600	1.04	1.37	0.65	0.94	2.2	2.1	
			700	0.96	0.97	0.21	0.88	3.1	2.7	
	BZY	30	600	0.93	1.34	0.82	0.89	2.2	1.9	
			700	0.84	1.13	0.29	0.80	2.7	2.1	
BSCF	BCZY1^/^	6	600	1.04	0.37	0.85	0.98	1.6	1.6	[71]
LSCF–BCZY3.5	BCZY3.5	8	600	1.05	0.31	1.25	0.99	2.6	2.6	[72]
			700	1.02	0.22	0.30	0.96	3.6	3.5	
LSCF–BSCZGY	BSCZGY	10	600	1.15	0.41	3.46	1.00	2.4	2.4	[73]
			700	1.13	0.19	1.82	1.00	5.4	5.4	
PBFM–SSC	BCZY1	25	600	1.01	0.41	0.54	0.95	6.1	5.8	[74]
NBFC	BCZD	30	600	1.05	0.68	0.66	0.96	4.4	4.2	[75]
			700	1.01	0.42	0.24	0.94	7.1	6.7	
NBFC’	BCZYY	25	600	1.04	1.04	0.83	0.95	2.4	2.3	[76]
			700	1.01	0.65	0.22	0.92	3.8	3.6	
PBC–BCZY	BCZY0.3	17	600	1.01	0.32	0.64	0.96	5.3	5.1	[77]
BFCC	BCZYY’	30	600	1.06	0.49	0.28	0.96	6.1	5.9	[78]
YBCZ	BCZD	20	600	1.03	0.77	0.51	0.94	2.6	2.5	[79]
			700	0.95	0.49	0.18	0.89	4.4	3.6	

^1^ In the most cases for the listed PCCs, the gas compositions represent wet (3%H_2_O) H_2_ and static (or wet) air. ^2^ Abbreviations of oxygen electrodes: PBN = Pr_1.9_Ba_0.1_NiO_4+δ_, SFM = SrFe_0.75_Mo_0.25_O_3−δ_, PLNCN = (Pr_0.9_La_0.1_)_2_Ni_0.74_Cu_0.21_Nb_0.05_O_4+δ_, PBC = PrBaCo_2_O_5+δ_, SSC = Sm_0.5_Sr_0.5_CoO_3−δ_, GBSC = GdBa_0.5_Sr_0.5_Co_2_O_5+δ_, NBFN = Nd_0.5_Ba_0.5_Fe_0.9_Ni_0.1_O_3−δ_, LSCF = La_0.6_Sr_0.4_Co_0.2_Fe_0.8_O_3−δ_, BSCF = Ba_0.5_Sr_0.5_Co_0.8_Fe_0.2_O_3−δ_, PBFM = (PrBa)_0.95_(Fe_0.9_Mo_0.1_)_2_O_5−δ_, NBFC = Nd_0.5_Ba_0.5_Fe_0.9_Co_0.1_O_3__−δ,_ NBFC’ = Nd_0.5_Ba_0.5_Fe_0.9_Cu_0.1_O_3__−δ_, PBC = PrBaCo_2_O_5+δ_, BFCC = BaFe_0.6_Co_0.3_Ce_0.1_O_3−δ_, YBCZ = YBaCo_3.5_Zn_0.5_O_7+δ_. ^3^ Abbreviations of electrolytes: BCZD = BaCe_0.5_Zr_0.3_Dy_0.2_O_3−δ_, BCZY1 = BaCe_0.7_Zr_0.1_Y_0.2_O_3−δ_, BCZY1^/^ = BaCe_0.8_Zr_0.1_Y_0.1_O_3−δ_, BCZY3 = BaCe_0.5_Zr_0.3_Y_0.2_O_3−δ_, BCZY3.5 = BaCe_0.5_Zr_0.35_Y_0.15_O_3−δ_, BCZY4 = BaZr_0.4_Ce_0.4_Y_0.2_O_3−δ_, BCZY6 = BaZr_0.6_Ce_0.2_Y_0.2_O_3−δ_, BCZY7 = BaZr_0.7_Ce_0.1_Y_0.2_O_3−δ_, BZCY8 = BaCe_0.1_Zr_0.8_Y_0.1_O_3−δ_, BZY = BaZr_0.8_Y_0.2_O_3−δ_, BSCZGY = Ba_0.5_Sr_0.5_Ce_0.6_Zr_0.2_Gd_0.1_Y_0.1_O_3−δ_, BCZYY = BaCe_0.5_Zr_0.3_Y_0.1_Yb_0.1_O_3−δ_, BCZYY’ = BaCe_0.7_Zr_0.1_Y_0.1_Yb_0.1_O_3−δ_, BCZYYC = BaCe_0.68_Zr_0.1_Y_0.1_Yb_0.1_Co_0.02_O_3−δ_.

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
