# Peer review of "A Reversible Protonic Ceramic Cell with Symmetrically Designed Pr2NiO4+δ-Based Electrodes: Fabrication and Electrochemical Features"

_materials, 2018, doi:10.3390/ma12010118_

Round 1

Reviewer 1 Report

The paper is a well-constructed work about the production and testing of reversible protonic ceramic cells (rPCCs) with symmetrical functional electrodes produced using a single sintering step. Experimental results are properly structured and functional characterizations were carried out to study the produced cell.

However, prior to publication, the authors may like to consider the following comments:

·         The one step sintering procedure was successfully applied for the production of PCC; this is one of the novelty of the paper so the author should stress more part the benefit of this production process in the introduction making an accurate review of the literature papers for single step Solid Oxide Cell development.

·         In the Results and Discussion part the porosity calculation with Image J and the EDS analyses should be done on the polished fracture to avoid inaccuracies in the data. Please put Figure 3 with an higher resolution, especially in the element distribution map.

·         In pag 4 and line 146 it is not clear why the porosity of functional electrode is measured using an as-prepared pellet  instead of measuring it by Image J in the real tape calendared layer. Consider that the presence of binder and the shaping method could influence the final porosity of the layer produced.

·         In the materials and method part, please put the cell dimensions

Author Response

Responses to Reviewers’ comments

We are grateful to the anonymous Reviewers for their extensive and valuable comments, which have allowed us to improve the overall quality of our work. We have addressed all the comments in the revised manuscript as discussed below. The corresponding changes in the revised manuscript are highlighted by red colour. Our response to each point raised by the Reviewers is given in blue font alongside the relevant comment.

Reviewer: 1

The paper is a well-constructed work about the production and testing of reversible protonic ceramic cells (rPCCs) with symmetrical functional electrodes produced using a single sintering step. Experimental results are properly structured and functional characterizations were carried out to study the produced cell.

However, prior to publication, the authors may like to consider the following comments:

1.     The one step sintering procedure was successfully applied for the production of PCC; this is one of the novelty of the paper so the author should stress more part the benefit of this production process in the introduction making an accurate review of the literature papers for single step Solid Oxide Cell development.

Response: The introduction section has been revised to highlight the relevance and novelty of the work:

According to the literature analysis, a single temperature processing step is a highly attractive approach for reducing the fabrication costs; in particular, this strategy has recently been adopted for the production of PCCs [8,22,23]. However, in these works the anode and cathode layers consisted of different functional materials, which can cause a mechanical misbalance leading to the deformation of whole cells following their sintering. Utilizing the same component for both functional electrode layers minimizes the possible strain, representing significant benefits in technological aspects, as well as the in terms of the quality of the target product”.

2.     In the Results and Discussion part the porosity calculation with Image J and the EDS analyses should be done on the polished fracture to avoid inaccuracies in the data. Please put Figure 3 with an higher resolution, especially in the element distribution map.

&

3.     In pag 4 and line 146 it is not clear why the porosity of functional electrode is measured using an as-prepared pellet instead of measuring it by Image J in the real tape calendared layer. Consider that the presence of binder and the shaping method could influence the final porosity of the layer produced.

Response: Unfortunately, at the present time, we do not have the possibility to repeat the SEM analysis on the polished samples. However, we are confident that the images (with high quality and contrast, with different magnifications) presented in the manuscript are of sufficient quality. Regarding the porosity determination, the Image J software was used to analyse the microstructure characteristics of the highly-porous supporting electrode, allowing measurement error to be estimated around 5%. For the functional electrodes having a lower porosity, the estimated error is proposed to be very high (indeed, in this case, analysis on the polished sections would be more suitable). Therefore, in order to give more accurate results, the porosity of the functional materials was determined on the individually prepared samples.

4. In the materials and method part, please put the cell dimensions

Response: The effective electrode area of the cell was mentioned in Section 2.4.

Reviewer 2 Report

In general paper is well written but there are several minor issues with English. See for example:

Line 119. The text reads As reported in Fig 2. Looks a bit weird. Can be changed to As it is shown.

Line 191. The text reads Impedance data and their analysis. Looks a bit weird. Can be changed to Impedance data and its analysis.

Other comments:

-EIS data in Fig 6b should be given with the same scale at both left and right y-axis.

-I recommend to authors to shorten a bit the manuscript by moving some part to supplementary information (For example Fig1 or XRD data).

-Figs 8b,9b. Why Y-axis labels is U,V but not power?

-Fig. 7. The fond size of title of x-axis is different from that on y-axis

-Fig 9a. Why title of x-axis is shifted from the center to left?

Author Response

Responses to Reviewers’ comments

We are grateful to the anonymous Reviewers for their extensive and valuable comments, which have allowed us to improve the overall quality of our work. We have addressed all the comments in the revised manuscript as discussed below. The corresponding changes in the revised manuscript are highlighted by red colour. Our response to each point raised by the Reviewers is given in blue font alongside the relevant comment.

Reviewer: 2

In general paper is well written but there are several minor issues with English. See for example:

1. Line 119. The text reads “As reported in Fig 2”. Looks a bit weird. Can be changed to “As it is shown”. Line 191. The text reads “Impedance data and their analysis”. Looks a bit weird. Can be changed to “Impedance data and its analysis”.

Response: The relevant corrections have been made.

Other comments:

2. -EIS data in Fig 6b should be given with the same scale at both left and right y-axis.

Response: The b panel was revised according to the Reviewer’s suggestion.

3. -I recommend to authors to shorten a bit the manuscript by moving some part to supplementary information (For example Fig1 or XRD data).

Response: We respectfully request that the current structure of the manuscript be maintained. We believe that we have reached an optimal combination between the Figures in the main text and those in the Appendix. Moreover, as can be seen from Point 12 of Reviewer 3, it is proposed that Figs. A1 – A5 be moved to the main text. Therefore, there are different points of view regarding this comment. At any rate, all the Figures will be presented in one file because no supplementary materials are used.

4. -Figs 8b,9b. Why Y-axis labels is “U,V” but not power?

5. -Fig. 7. The fond size of title of x-axis is different from that on y-axis

6. -Fig 9a. Why title of x-axis is shifted from the center to left?

Response: We thank you for noticing our omission. The corresponding corrections have been made.

Reviewer 3 Report

The paper “A Reversible Protonic Ceramic Cell with Symmetrically Designed Pr2NiO4+δ-Based Electrodes: Fabrication and Electrochemical Features” demonstrates the feasibility of using PNO-based electrode for symmetrical designed reversible PCC. This paper is well written, and full of information. Some issues need to be addressed before its publication.

1.       Experimental: Page 2 Line 69, BPN powders synthesis can not be found in the reference that authors provided. Please specify the synthesis method in this paper.

2.       Page 3 Line 105, please provide references for DRTTools.

3.       Page 4 Line 129-131, from the room temperature XRD, the transformation from P2O3 to Pr(OH)3 can not be proved. Instead, a high temperature XRD must be implemented for drawing this conclusion.

4.       Figure 2a, why is XRD for (Nd,Ba)2NiO4 shown in the figure? Did authors do any XRD on PBN powder after calcination?

5.       Page 4, Line 140, what is Ln in this case? Please specify.

6.       Page 6 Line 177, the maximum powder densities are not “high”, the authors need to replace the word.

7.       Figure 4d is unnecessary, because the information can be simply seen in figure 4b and 4c.

8.       Figure 8d caption, do you mean “depending on pH2O”? Also, for Figure 8d, the x-axis should b e spaced corresponding to pH2O rather than equally spaced.

9.       Page 16 Line 414-415, how do you measure performance under OCV condition? Also, “fuel cell mode” instead of “fuel mode”?

10.   In Abstract, the authors mention “The electrolysis mode of the rPCC is found to be more efficient than the fuel cell mode under highly humidified atmosphere”. Which side is humidified in this case? What is the measure of efficiency. What is the condition for comparison? I could not find direct comparison of fuel cell and electrolysis mode in the full text.

11.   In all DRT analysis, it is difficult to keep track of all the physical meaning of all the different peaks. Can the author indicate the physical meanings of all the peaks? Otherwise, the results from DRT will be useless.

12.   All the Appendix data have to be moved to the main text, because they are as important as those presented in the main text.

Author Response

Responses to Reviewers’ comments

We are grateful to the anonymous Reviewers for their extensive and valuable comments, which have allowed us to improve the overall quality of our work. We have addressed all the comments in the revised manuscript as discussed below. The corresponding changes in the revised manuscript are highlighted by red colour. Our response to each point raised by the Reviewers is given in blue font alongside the relevant comment.

Reviewer: 3

The paper “A Reversible Protonic Ceramic Cell with Symmetrically Designed Pr2NiO4+δ-Based Electrodes: Fabrication and Electrochemical Features” demonstrates the feasibility of using PNO-based electrode for symmetrical designed reversible PCC. This paper is well written, and full of information. Some issues need to be addressed before its publication.

1. Experimental: Page 2 Line 69, BPN powders synthesis can not be found in the reference that authors provided. Please specify the synthesis method in this paper.

Response: Please accept our apologies for the incorrect link. In the revised manuscript, the corresponding reference is correct and contains the relevant details of the synthesis method (see page 2).

2. Page 3 Line 105, please provide references for DRTTools.

Response: The link to the corresponding website for this software has been added.

3. Page 4 Line 129-131, from the room temperature XRD, the transformation from P2O3 to Pr(OH)3 cannot be proved. Instead, a high temperature XRD must be implemented for drawing this conclusion.

Response: We have provided confirmation of our conclusion:

This is confirmed by the fact that the total weight change of PBN under full reduction is equal to 95.2% (δ@RT = 0.17, 4 + δ@1000 °C = 2.95; see Figure 2b) corresponding with the formation of 0.95 mole of Pr2O3 and 0.1 mole of BaO at 1000 °C. Otherwise, if Pr(OH)3 was formed during the reduction procedure, the overall weight change should amount to 96.1%. This level corresponds to δ@RT = –0.07 (4 + δ@1000 °C = 2.95), which is in disagreement with that reached for Pr2NiO4+δ, which has a close composition, δ = 4.23–4.25 [32,33]”.

4. Figure 2a, why is XRD for (Nd,Ba)2NiO4 shown in the figure? Did authors do any XRD on PBN powder after calcination?

Response: Thank you for noticing our typo. We have replaced it with (Pr,Ba)2NiO4.

5. Page 4, Line 140, what is Ln in this case? Please specify.

6. Page 6 Line 177, the maximum powder densities are not “high”, the authors need to replace the word.

Response: The corresponding corrections have been made.

7. Figure 4d is unnecessary, because the information can be simply seen in figure 4b and 4c.

Response: Figs 4, 8 and 9 were presented in the same manner. Regardless the fact that the target values can be obtained from panels (b) and (c), we provided the histograms (d) for clearer analysis of the results.

8. Figure 8d caption, do you mean “depending on pH2O”? Also, for Figure 8d, the x-axis should be spaced corresponding to pH2O rather than equally spaced.

Response: We thank you for noticing our omission. The corresponding correction has been made.

9. Page 16 Line 414-415, how do you measure performance under OCV condition? Also, “fuel cell mode” instead of “fuel mode”?

Response: These parts of the test have been clarified.

10. In Abstract, the authors mention “The electrolysis mode of the rPCC is found to be more efficient than the fuel cell mode under highly humidified atmosphere”. Which side is humidified in this case? What is the measure of efficiency? What is the condition for comparison? I could not find direct comparison of fuel cell and electrolysis mode in the full text.

Response: Panel (d) of Fig. 9 shows the performance response to the humidification of both atmospheres. As can be seen, such a humidification results in a deterioration of Pmax, but an improvement of jH2. The term “efficient” was replaced with “appropriate”, since the former term was associated in the manuscript with ion transference numbers.

11. In all DRT analysis, it is difficult to keep track of all the physical meaning of all the different peaks. Can the author indicate the physical meanings of all the peaks? Otherwise, the results from DRT will be useless.

Response: The analysis of DRT data (as well as impedance data and equivalent circuits) is a difficult task, which is usually based on the concept of the probability of occurring processes. In other words, during a certain mode of operation, more than 10 elementary reactions occur (see Table); therefore their number might be even more dependent on transport behaviors of electrolyte and electrode systems. Therefore, we cannot provide the exact physical meaning of each peak (arc, semicircle). However, on the base of capacitance and frequency values, it is possible to make conclusions regarding the nature of ongoing process (adsorption, diffusion, charge transfer). This was done when discussing the DRT data (Figs 7, 10 and A3).

Elementary steps realised at PCEC electrodes under water electrolysis.

Electrode

Elementary reaction

Description

No.

anode

← frequency

incensement

water adsorption

1

water dissociation

2

adsorbed spaces dissociation

3

oxygen oxidation

4

oxygen oxidation

5

oxygen association

6

molecular oxygen formation

7

proton diffusion to TPB

8

proton incorporation in lattice

9

cathode

frequency →

incensement

proton diffusion to electrolyte surface

10

proton transfer through interface

11

hydrogen reduction

12

molecular hydrogen formation

13

12.   All the Appendix data have to be moved to the main text, because they are as important as those presented in the main text.

Response: We request that the current structure of the manuscript be maintained. We believe that we have reached an optimal combination between the Figures in the main text and those presented in the Appendix. Moreover, as can be seen from the Point 3 of Reviewer 2, some Figures from the main text are proposed to be moved to the Appendix. Therefore, there are different points of view regarding this comment. Besides, all the Figures will be presented in one file because no supplementary materials are used.

Round 2

Reviewer 1 Report

Response of point 3: the SEM images are of good quality but for the porosity determination with software analysis is better the use of polished fracture to have a more representative section of sample. However the overall work is good and suitable for publication